# Impact of Time Point of Extracorporeal Membrane Oxygenation on Mortality and Morbidity in Congenital Diaphragmatic Hernia: A Single-Center Case Series

**DOI:** 10.3390/children9070986

**Published:** 2022-07-01

**Authors:** Christian Wegele, Yannick Schreiner, Alba Perez Ortiz, Svetlana Hetjens, Christiane Otto, Michael Boettcher, Thomas Schaible, Neysan Rafat

**Affiliations:** 1Department of Neonatology, University Children’s Hospital Mannheim, University of Heidelberg, 68167 Mannheim, Germany; christianwegele@web.de (C.W.); yschreiner@gmx.de (Y.S.); alba.perez-ortiz@umm.de (A.P.O.); thomas.schaible@umm.de (T.S.); 2Department of Neonatology, Pediatric Intensive Care and Sleep Medicine, Vestische Kinder-Jugendklinik Datteln, University Witten/Herdecke, 45711 Datteln, Germany; 3Department of Biomathematics and Medical Statistics, Medical Faculty Mannheim, University of Heidelberg, 68167 Mannheim, Germany; svetlana.hetjens@medma.uni-heidelberg.de; 4Department of Obstetrics and Gynecology, University Medical Center Mannheim, University of Heidelberg, 68167 Mannheim, Germany; christiane.otto@umm.de; 5Department of Pediatric Surgery, University Children’s Hospital Mannheim, University of Heidelberg, 68167 Mannheim, Germany; michael.boettcher@umm.de

**Keywords:** congenital diaphragmatic hernia, extracorporeal membrane oxygenation, chronic lung disease

## Abstract

Since there are no data available on the influence of the time point of ECMO initiation on morbidity and mortality in patients with congenital diaphragmatic hernia (CDH), we investigated whether early initiation of ECMO after birth is associated with a beneficial outcome in severe forms of CDH. All neonates with CDH admitted to our institution between 2010 until 2020 and undergoing ECMO treatment were included in this study and divided into four different groups: (1) ECMO initiation < 12 h after birth (*n* = 143), (2) ECMO initiation between 12–24 h after birth (*n* = 31), (3) ECMO initiation between 24–120 h after birth (*n* = 48) and (4) ECMO initiation > 120 h after birth (*n* = 14). The mortality rate in the first (34%) and fourth group (43%) was high and in the second group (23%) and third group (12%) rather low. The morbidity, characterized by chronic lung disease (CLD), did not differ significantly in the three groups; only patients in which ECMO was initiated >120 h after birth had an increased rate of severe CLD. Our data, although not randomized and limited due to small study groups, suggest that very early need for ECMO and ECMO initiation > 120 h after birth is associated with increased mortality.

## 1. Introduction

Congenital diaphragmatic hernia (CDH) is characterized by the failure of diaphragmatic development and prenatal thoracic herniation of abdominal organs, leading to lung hypoplasia and persistent pulmonary hypertension of the newborn (PPHN) [1]. If conventional treatment with gentle ventilation and optimized vasoactive medication fails, extracorporeal membrane oxygenation (ECMO) may be considered [1]. Currently, CDH is the most common indication for ECMO in neonates [2], but survival rates reported by the extracorporeal life support organization (ELSO) have continued to drop in the modern era [3]. Indications for initiating ECMO include either respiratory or circulatory parameters, which are also undergoing continuous refinement. The benefits of ECMO in CDH are still controversial, and systematic reviews concerning a benefit of ECMO in CDH did not find an advantage for this therapeutic option [4,5,6]. However, some centers and networks have demonstrated an increase in survival rates in CDH with the employment of ECMO by retrospective analysis in their series [7,8]. However, data on long-term results, including morbidity and quality of life, are rare. Survival might be influenced by the timing of ECMO initiation and the timing of surgical repair. In this regard, a trend toward early initiation of ECMO and early surgery on ECMO exists [2]. The use of ECMO in CDH will continue to be evaluated, and prospective randomized trials and registry network are necessary to help in answering the addressed questions of patient selection and management.

We hypothesized that an early initiation of ECMO after birth is associated with a beneficial outcome in severe forms of CDH. In this study, we evaluated the impact of the timing of ECMO initiation on the clinical outcome of CDH patients. 

## 2. Materials and Methods

### 2.1. Subjects and Clinical Data

All newborn infants with CDH treated between 1 January 2010 and 31 March 2020, who received ECMO support at our neonatal intensive care unit (NICU) at the Department of Neonatology of the University Children’s Hospital Mannheim, University of Heidelberg, were included in this retrospective study. ECMO support was initiated according to the recommendations suggested by the CDH EURO Consortium Consensus and ELSO [2,9]. Exclusion criteria were patients with associated anomalies (including cardiac malformation), syndromes or chromosomal aberrations. The study population was divided into the four following groups depending on the time point of ECMO initiation: (1) ECMO initiation < 12 h after birth, (2) ECMO initiation between 12–24 h after birth, (3) ECMO initiation between 24–120 h after birth and (4) ECMO initiation 120 h after birth. The surgical correction was performed after stabilization of the neonate after ECMO in all cases. Pre-, peri- and postnatal clinical parameters were collected from the patient’s records, and mortality and morbidity were compared between the groups. Morbidity was represented by duration of mechanical ventilation, duration of hospitalization and development of chronic lung disease (CLD). The diagnosis of CLD was made as reported before [10,11]: if there was an additional need for oxygenation at day 28 after birth, CLD was diagnosed. Severity of CLD was differentiated into three grades according to the additional need for oxygenation at day 56 after birth: mild CLD with no need for supplemental inspired oxygen (fraction of inspired oxygen (Fio2) ≤ 0.21), moderate CLD (Fio2, 0.22–0.29), and severe CLD (Fio2 ≥ 0.30). For some patients being discharged to other hospitals before the 56th day of life, CLD could be diagnosed, but severity could not be assessed. If outcome information was missing, patients were categorized as ‘CLD n/a’. The defect size of each CDH patient was assessed according to the consensus of the CDH study group [12]. This study was approved by the local ethics committee of the Medical Faculty Mannheim of the University of Heidelberg (reference number: 2019-884R).

### 2.2. Statistical Analysis

Statistical calculations were performed using SAS software, release 9.4 (SAS Institute Inc., Cary, NC, USA). For qualitative variables, absolute and relative frequencies are given. For quantitative and approximately normally distributed variables, mean values and standard deviations were calculated. For skewed or ordinal data, minimum and maximum are presented. To compare groups regarding qualitative parameters, a Chi-square test or Fisher’s exact test was used where appropriate. For normally distributed data, a one-way analysis of variance (ANOVA) was performed to compare the mean values of different age groups. For these analyses, the SAS procedure PROC MIXED was used. Adjustment for multiple comparisons was done by Scheffé test. A *p*-value of <0.05 was considered statistically significant.

## 3. Results

In this study, 243 patients were included. One patient was excluded because of missing data about the start of the ECMO and six patients were excluded for additional cardiac malformation. For an overview of the recruitment of CDH patients into this study and the characteristics of the dropouts, please see Appendix A. 

Of 236 analyzed patients, 145 (61.4%) were male and 91 (38.6%) were female. Further, 188 (79.7) patients presented with left-sided and 48 (20.3%) patients with right-sided CDH. Moreover, 209 (88.6%) children were diagnosed with CDH prenatally, whereas 27 (12.4%) were still unknown on delivery. For an overview of the characteristics of the study population, please refer to Table 1.

Overall survival was 71.6%, but when looking at each group individually, survival rates differed from 66.4% (*n* = 95) in the first group to 77% (*n* = 24) in the second group, 87.5% (*n* = 42) in the third group and 57% (*n* = 8) in the fourth group, respectively (Table 2). Survival in group 2 and group 3 was significantly higher than in group 1 and group 4 (*p* = 0.011). 

Some clinical data were collected and compared for all the groups (Table 2). Values for AaDO_2_ prior to ECMO initiation for group 1 versus group 2 (*p* = 0.035) and group 1 versus group 3 (*p* < 0.0001) were significant (Table 2). Further, liver herniation was significantly less in patients in group 3 and group 4 (*p* < 0.001) (Table 2). For the remaining clinical parameters, differences between the groups were not significant (Table 2). 

When looking at parameters for ventilation and oxygenation prior to ECMO initiation, the highest pCO_2_ was determined in group 1, requiring ECMO support earliest compared to the other groups. Levels of pCO_2_ immediately before ECMO initiation in group 1 were significantly higher compared to group 3 (*p* < 0.0001) and marginally higher compared to group 4, respectively (*p* = 0.052) (Figure 1A). There were no significant differences between the groups concerning the level of paO_2_ prior to ECMO initiation (Figure 1B). 

Evaluation of morbidity was performed by assessing chronic lung disease (CLD) and for this purpose, patients from group 1 and group 2 were pooled together due to the need of ECMO for the same pathophysiological reason, namely pulmonary ventilation disorder. In group 1 and 2, the incidence of CLD was significantly lower compared to the other two groups (Figure 2). CDH patients in group 3 had the highest number of mild CLD cases, whereas patients in group 4 had significantly higher numbers of moderate and severe CLD (Figure 2). 

The defect size of each CDH patient was assessed according to the consensus of the CDH study group. There was no significant difference regarding defect size between the four different groups (Table 3). 

## 4. Discussion

In this study, we collected clinical parameters over a ten-year period from CDH patients undergoing ECMO, which were referred to our center. There is evidence suggesting a critical time point for ECMO initiation in patients with CDH to improve outcome. To our knowledge, this is the first report to show that CDH newborns with a later ECMO initiation, between 12 h and 120 h after birth, have a significant survival benefit compared to newborns undergoing a very early or very delayed ECMO intervention within the first 12 h or beyond 120 h of life, respectively. 

### 4.1. Indication for ECMO in CDH

ECMO in neonates should be initiated when indication occurs. Indication in classical neonatal ECMO (e.g., meconium aspiration syndrome) is usually present when oxygenation index is above 40 [2,13]. However, in CDH, indication for ECMO is more complex [1,9]. While applying gentle ventilation for small lungs, the mean airway pressure does not exceed values over 15. Therefore, with an FiO_2_ of 1.0 and arterial paO_2_ below 50mmHg, where hypoxia starts to occur, OI is only 30 and may be a cut off to start ECMO in CDH [1,9]. 

However, in most cases, hypercarbia, not hypoxia, and metabolic acidosis trigger severe pulmonary hypertension (PHT) and an entry criterion in CDH may be a persisting pH-value below 7.15 (corresponding with paCO_2_ values between 60 and 100 mmHg) [1,9]. In such a situation of acidosis, PHT can also be recognized as a pre-postductal saturation difference of more than ten percent. 

A third indication for starting ECMO is circulatory failure, with low cardiac output, low urinary output and rising lactate [1,2,9]. However, this criterion is rare and often occurs later than hypercapnia and hypoxia. A honeymoon period occurs in those patients, in which a respiratory acidosis can be overwhelmed and pCO_2_-value below 60 mmHg can be reached.

### 4.2. Parameters for Predicting ECMO

There are many data about prenatal parameters and prognosis [14]. To date, LHR, TFLV and liver herniation into the thoracic cavity are the most reliable prenatal predictors for mortality in CDH [15]. Likewise, liver herniation and lung volume could also predict the need for ECMO support in these patients [10]. However, the prognostic value of prenatal data is not clear and treating patients with the same disease severity may lead to different results and, therefore, it is difficult to compare treatment centers. Treatment strategies may vary even between high-volume centers and results are not the same [7]. 

Significantly fewer data are available for prognostic value of early postnatal parameters predicting mortality [16,17]. Of course, high OI calculated by respiratory parameters after delivery leads more often to ECMO than lower OI, and survival of CDH patients with ECMO is lower than survival of CDH patients without ECMO [18]. For example, Wilford Hall Santa Rosa Formula (highest paO_2_—highest paCO_2_ during initial 24 h of life) includes values of pCO_2_ [16,19]. If the difference of the best paO_2_ minus highest paCO_2_ is below zero, there is high risk for mortality. Postnatal prognostic values of this formula are fairly high and calculation is possible as early as one hour after delivery and standardized primary care. Applying the Wilford Hall Santa Rosa Formula to our results, the value is lowest in group 1 with the poorest prognosis.

### 4.3. Honeymoon in CDH Prior to ECMO

A honeymoon period with higher values of oxygenation (temporarily arterial paO_2_ over 60 mmHg) and vanishing pre-postductal saturation difference occurs in those patients in whom a respiratory acidosis can be overwhelmed and pCO_2_-value below 60 mmHg can be reached. A honeymoon period may indicate a better responsiveness of pulmonary vessels and, therefore, a better prognosis. This scenario mostly occurs in patients from group 2 and 3 and leads to ECMO success rates of about 80%. The lower paO_2_ directly before initiation of ECMO reflects the sudden end of the honeymoon period with a rebound phenomenon caused by vasoconstriction of pulmonary vessels.

Patients without a honeymoon period do not respond to treatment strategies prior to ECMO and are treated with ECMO earlier and this course leads to a poorer prognosis, with survival rates from ECMO of 66% in group 1. Looking at the basic data of these patients, they started even prenatally with the lowest values of fetal lung volume. Surprisingly, the outcome of initial honeymooners, but late decision for ECMO because of a longer apparently stable phase of several days, was poorest. This result leads one to speculate that responsiveness of pulmonary vascular bed to the unloading effect of ECMO and, therefore, the potential to overwhelm PHT in CDH decreases towards the end of the first week of life. It can be assumed that the effectiveness of the PHT treatment has become blunted.

Reaching a honeymoon by high-frequency oscillation ventilation (HFOV) may occur in some patients but one side effect of HFOV is a tendency to overdistension of the lungs and the effort of controlling paCO_2_ often leads to hypoxemia. The VICI trial showed that HFOV ventilation seems inferior to CMV [20]. HFOV is able to control paCO_2_ but may delay ECMO initiation and, therefore, poorer survival rates were seen in CDH patients treated with HFOV as the initial mode of ventilation [20]. 

### 4.4. Chronic Lung Disease after ECMO

The development of CLD represents an important risk factor for impaired pulmonary outcome in CDH patients. Although neonatal ECMO has improved the survival of CDH patients, improved survival might carry a higher risk of long-term morbidity among survivors [21]. There is conflicting evidence about the development of CLD in ECMO survivors. In an older prospective study, ECMO support resulted in a 50% reduction in CLD in survivors of severe respiratory failure [22]. Other studies have demonstrated that ECMO support does not prevent sequelae of severe respiratory disease in the newborn period [23]. Since lung development is already disturbed in CDH due to lung hypoplasia and pulmonary hypertension, additional risk factors, such as CLD, impair pulmonary outcome significantly.

Our study adds data concerning the pulmonary outcome defined by the severity of CLD. The risk for developing CLD is very high in group 3 and 4, and extremely high for developing severe CLD in group 4 (50%). Although patients in group 4 had a seemingly better prenatal expected prognosis in comparison to group 3 due to, e.g., less liver-up, mortality and morbidity (severe CLD) were much higher in this group. Since early ECMO initiation was associated with less mortality and severe CLD, it appears that ECMO should be started earlier, independently from the indication’s parameter.

Whether persisting pulmonary hypertension or more severe lung damage with longer time on high ventilator settings are more contributable to CLD was not investigated.

## 5. Conclusions

Our data, although not randomized and limited due to small study groups, suggest that very early need for ECMO and ECMO initiation >120 h after birth is associated with increased mortality and even with morbidity in late ECMO initiation. Therefore, future studies have to be conducted to identify and stratify the best timepoint for initiating ECMO in the different forms of CDH, depending on prenatal risk factors for ECMO employment (e.g., fetal lung volume, liver-up, etc.) and postnatal factors (e.g., degree of PHT, acidosis, etc.).

## Figures and Tables

**Figure 1 children-09-00986-f001:**
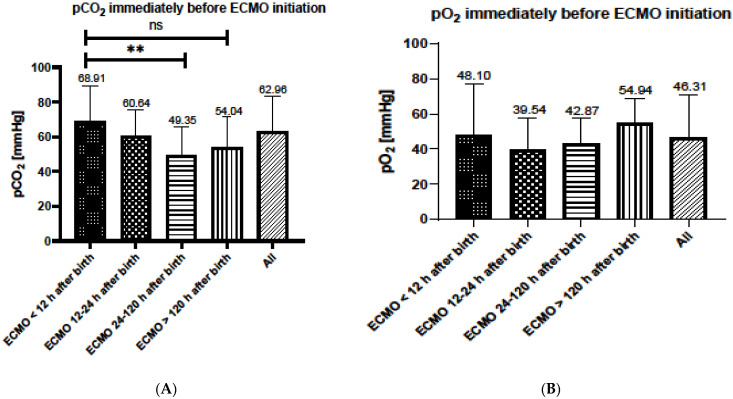
Ventilation (**A**) and oxygenation (**B**) parameters prior to ECMO initiation. ** statistically significant. “ns” stands for not significant.

**Figure 2 children-09-00986-f002:**
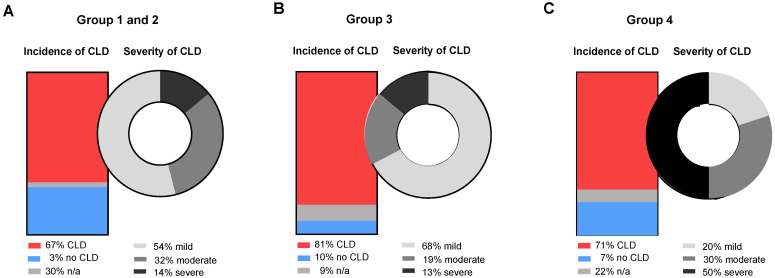
Incidence and severity of chronic lung disease (CLD) in the different groups of the study population: (**A**) group 1 (ECMO initiation <12 h after birth) and 2 (ECMO initiation 12–24 h after birth), (**B)** group 3 (ECMO initi ation 24–120 h after birth) and (**C**) group 4 (ECMO initiation 120 h after birth).

**Table 1 children-09-00986-t001:** Clinical data for each study group.

	Study Group									
	1 (*n* = 143)		2 (*n* = 31)		3 (*n* = 48)		4 (*n* = 14)		All (*n* = 236)	
GA (weeks)	37.3	±1.40	38.1	±1.50	38.0	± 1.68	38.0	± 0.68	37.6	±1.47
GW (g)	2915	±422	3140	±535	3035	±624	3015	±552	2975	±494
ApH	7.31	±0.1	7.30	±0.1	7.30	±0.08	7.25	±0.11	7.30	±0.07
5 min APGAR	7.31	±1.20	7.30	±1.90	7.30	±1.77	7.25	±1.16	7.07	±1.46
paO_2_ 6 h prior to ECMO (mmHg)	72.9	±73.0	65.3	±34.0	79.4	±53.2	77.8	±51.0	74.3	±56.1
Age at ECMO initiation (h)	5.68	±2.00	17.5	±3.60	48.1	±25.5	240	±123	29.7	±63.3
Duration of ECMO (h)	247	±105	247	±99.1	207	±76.9	208	±78.1	237	±98.6
AaDO_2_ prior to ECMO	596	±33.1	613	±17.4	621	±18.4	604	±22.2	604	±29.9
OI prior to ECMO	36.1	±20.4	43.6	±28.6	35.6	±17.7	27.9	±14.8	36.5	±21.0
Duration of MV (d)	35.2	±33.2	36.2	±20.5	37.6	±22.9	41.4	±21.0	36.2	±29.2
Duration of hospitatlization (d)	73.5	±62.5	84.6	±70.5	90.1	±63.3	67.2	±44.8	77.7	±62.6
TFLV (%)	27.4	±10.8	28.2	±11.6	34.4	±12.8	30.6	±7.52	28.8	±11.2
o/e LHR (%)	35.0	±11.4	41.7	±20.6	37.0	±10.2	37.8	±14.1	36.2	±12.5

Data are presented in means ± standard deviation (SD). GA = gestational age, GW = gestational weight, ApH = arterial umbilical cord pH, ECMO = extracorporeal membrane oxygenation, OI = oxygenation index, TFLV = total fetal lung volume, o/e LHR = observed/expected lung-to-head-ratio.

**Table 2 children-09-00986-t002:** Survival and frequencies of CDH patch, abdominal patch and liver-up phenomenon.

	Study Group				*p*-Value
	1 (*n* = 143)	2 (*n* = 31)	3 (*n* = 48)	4 (*n* = 14)	
Survival	0.66	0.77	0.88	0.57	**0.0110**
CDH patch	0.99	1.00	0.96	1.00	0.34
Abdominal patch	0.45	0.32	0.30	0.36	0.31
Liver-up	0.93	0.93	0.74	0.69	**0.0009**

Data are presented in percentage of each group’s study population. CDH = congenital diaphragmatic hernia. The bold points out that these *p*-values are statistically significant.

**Table 3 children-09-00986-t003:** Frequencies of CDH defect size (A–D) according to CDH study group consensus.

		A	B	C	D	*p*-Value
Group	1 (*n* = 143)	0.01	0.09	0.65	0.25	0.46
	2 (*n* = 31)	0.00	0.16	0.73	0.11	
	3 (*n* = 48)	0.00	0.26	0.55	0.19	
	4 (*n* = 14)	0.00	0.14	0.72	0.14	

## Data Availability

The data presented in this study are available on request from the corresponding author.

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
