# Peer review of "Impact of Time Point of Extracorporeal Membrane Oxygenation on Mortality and Morbidity in Congenital Diaphragmatic Hernia: A Single-Center Case Series"

_children, 2022, doi:10.3390/children9070986_

Round 1

Reviewer 1 Report

This article, titled "Impact of time point of extracorporeal membrane oxygenation 2 on mortality

and morbidity in congenital diaphragmatic hernia." reports a ten-year single-center experience with

CDH patients, who had undergone an ECMO treatment. The authors stratified the cohort of patients

according to the timing after birth when the ECMO treatment was initiated, showing that the very

early and the very delayed start of this treatment negatively influence the prognosis.

Here are my comments:

• The title is clear. I suggest mentioning that it is a case series and a single-center experience.

• The abstract clearly summarizes the backgrounds, the methods, and the results of the study.

• Introduction: In the first sentence I suggest specifying that the error occurs in the prenatal

period (e.g., “….development and prenatal thoracic herniation of abdominal….”). For the rest, I think

the introduction is clear and complete.

• In the methods: The criteria for initiating ECMO are not specified. Furthermore, it is not clear

whether these criteria underwent some modification during the study period. If it is the case, I

suggest specifying this, if not the authors should mention the differences.

• In figure 2: I suggest improving the rounding of the percentage values so that the sum is

100% (e.g., Fig 2. Group 1 and 2: Incidence: 67% CLD; 3% no CLD; 29% n/a = 99%).

• In the results: I suppose the determination of the defect was made during surgery. However,

In the manuscript, I do not see any mention of the timing of the surgical correction. Did patients in

group 1 all receive the correction before ECMO? How many patients passed away before surgery?

Was in these cases the defect size determined during the autopsy? The authors should in my opinion

give some complementary information.

• In Table 3: I suggest improving the rounding of the percentage values so that the sum is 1.

• In the discussion the authors state: “This may suggest that early ECMO initiation may not

only result in less mortality but protect from CLD.” I suggest reformulating this sentence. It appears

in such form that ECMO should be started earlier, independently from the indication’s parameters.

The manuscript is well structured and comprehensible. The information given represents an

an important contribution to improving knowledge about the role of ECMO in treating patients

affected by severe forms of CDH.

In conclusion, after the suggested minor improvements, I recommend the manuscript, for

publication in Children.

Author Response

We thank the reviewer for taking the time to read our manuscript and his/her thoughtful comments:

To 1.) We have edited the title and added “…: a single-center case series”

To 2.) We have added “prenatal” as suggested in the first sentence of the introduction.

To 3.) We have added in the methods section the Criteria for initiating ECMO and cited the guidelines of the CDH EURO Consortium Consensus and ELSO. There was no modification in this regard.

To 4.) We have edited the numbers in Figure 2 according to the suggestions.

To 5.) Yes, the defect size is determined during surgery. At our center, the surgical correction is only performed after the stabilization of the neonate after ECMO. All of the included patients had surgical correction.

We have added a comment in the methods section referring to this.

To 6.) We have edited the numbers in Table 3 according to the suggestions.

To 7.) We have reformulated the sentence according to the suggestions.

Reviewer 2 Report

First of all, I want to note that it has been a pleasure review your manuscript. I think this is an interesting topic for clinicians who manage this severe condition.

Interesting work on the evaluation of the efficacy of ECMO in patients with diaphragmatic hernia on morbidity and mortality.

After reading in depth the manuscript, I would like to make some comments and ask the authors several questions about.

- There are lines on different pages that do not end correctly. For example 50,62, 93...we suggest to review the whole document.

- In the supplementary material, figure 1, it comments on "malformations" in a general way, in which it is assumed that cardiac malformations would also be included, however, below cardiac malformations are identified. This should be better specified to avoid confusion.

- line 101 is a little tapped

- The number of patients was high but the sample size could have been calculated to have a good statistical power.

- In each study group, the sample sizes differed greatly. It is then commented that it depended on certain parameters...such as the partial pressure of carbon dioxide...but could this problem have been solved so that the samples were more homogeneous in terms of the number of subjects evaluated?

- It should have been made clear beforehand which parameters were necessary for the patient to be administered ECMO before or after. It is commented in the results, however it could have been specified earlier in the material and methods section, when explaining the division of the 4 groups.

- Line 129 larger font size.

- There are some references in which several authors are referenced. Explanation should be given.

- Reference number 1 is missing page

- Some references are in capital letters.

Author Response

We thank the reviewer for taking the time to read our manuscript and his/her thoughtful comments as well as his kind introducing statement:

To 1.) We have edited the document accordingly.

To 2.) We agree that the listing of “malformations” and “cardiac malformations” can be confusing for the reader. We just wanted to show our complete numbers and out of the 628 CDH patients treated at our centers in this time period, 385 patients did not receive ECMO, because 328 patients didn’t need it and 57 patients weren’t even offered ECMO due to associated malformations (including cardiac malformations). The second group of patients which have been excluded all received ECMO, but for one patient we had a missing exact time point, because ECMO was initiated in a different center, before the patients was referred and transport (undergoing ECMO) to us. The other 6 patients had additional cardiac malformation, therefore we excluded them for our analysis to have a rather homogenous group of CDH.

For some kind of clarity, we have edited the term “cardiac malformation” to “congenital heart defect”.

To 3.) Line 101 has been edited.

To 4.) We agree that a power analysis could have been calculated, but for retrospective studies this is always tricky. We have added all the patients from our center from the time point (2010) when ECMO initiation for CDH was standardized at our center according to the CDH European Consortium Consenses.

To 5.) This is due to the fact that this is a retrospective study and we defined the group according to certain timepoints, which in our opinion/experience make sense to evaluate. We do not see a reasonable way to have more homogenous sample sizes per group.

To 6.) We have added in the methods section the criteria for initiating ECMO and cited the guidelines of the CDH EURO Consortium Consensus and ELSO. The details (including the parameters) can be found in the citations.

To 7.) The font size has been edited.

To 8.) You wrote: “There are some references in which several authors are referenced. Explanation should be given.” Unfortunately, we have not exactly understood, what the reviewer intended to tell us. Can you please explain what you exactly mean? Please excuse this inconvenience.

To 9.) This reference has no page numbers, because Frontiers in Pediatrics is a purely Online Journal.

To 10.) In the revised manuscript all the references have been edited accordingly.